# Examining the prevalence and determinants of early initiation of breastfeeding: Evidence from the 2017/2018 Benin demographic and health survey

Ebenezer Kwesi Armah-Ansah[1,2,3]*, Elvis Ato Wilson[4], Kenneth Fosu Oteng[5], Benedicta Bawa[6], Joseph Yaw Dawson[7]

1 Department of Population and Health, University of Cape Coast, Cape Coast, Ghana, 2 Population Dynamics Sexual and Reproductive Health Unit, African Population and Health Research Center, Nairobi, Kenya, 3 Department of Population and Development, National Research University–Higher School of Economics, Moscow, Russia, 4 Kintampo Health Research Centre, Ghana Health Service, Kintampo, Ghana, 5 Ashanti Regional Health Directorate, Ghana Health Service, Kumasi, Ghana, 6 One Trust LTD, East Legon, Accra, Ghana, 7 Ural Federation University, Yekaterinburg, Russia

* ebenezer.armah-ansah@stu.ucc.edu.gh

**Data Availability Statement:** Data is available on https://dhsprogram.com/publications/publication-FR350-DHS-Final-Reports.cfm.

## Abstract

Early initiation of breastfeeding has been noted as one of the well-known and successful interventions that contributes to the reduction of early childhood mortality and morbidity. The Government of Benin has established multi-sectoral institutions and policies to increase the prevalence of early initiation of breastfeeding. However, there is little information on the prevalence and the determinants of early initiation of breastfeeding in Benin. This study therefore sought to examine the prevalence and determinants of early initiation of breastfeeding among women in Benin. This is a secondary data analysis of the 2017/2018 Benin demographic and health survey. The study included weighted sample of 7,223 women between the ages of 15 and 49. STATA was used for the data analysis. We used a multilevel logistic regression to investigate the factors of early breastfeeding initiation in Benin. To determine the significant relationships, the data were reported as odds ratios (ORs) with 95% confidence intervals (CIs) and p-value 0.05. The prevalence of early initiation of breast-feeding among mothers was 56.0%. Early initiation of breastfeeding was lower among employed women (aOR = 0.80, 95% CI = 0.69–0.94), women who had caesarean section (aOR = 0.21, 95% CI = 0.16–0.28), those exposed to mass media (aOR = 0.85, 95% CI = 0.75–0.96) and women who received assistance at birth from skilled worker (aOR = 0.57, 95% CI = 0.46–0.71). The findings of this study showed that four in ten children miss early initiation of breastfeeding in Benin. The findings, therefore, call for the need for policymakers to shape existing programs and consider new programs and policies to help improve early initiation of breastfeeding practices in Benin. It is, therefore, recommended that information, education and communication programs targeting mothers who are less likely to practice early initiation of breastfeeding be formulated, implemented, and monitored accordingly by the Ministry of Health.

**Funding:** the author(s) received no specific funding for this work.

**Competing interests:** The authors have declared that no competing interests exist.

**Abbreviations:** WHO, World Health Organization; EIBF, Early Initiation of Breastfeeding; ANC, Antenatal care; LMICs, Low-and-middle-income countries; SSA, Sub-Saharan Africa; BDHS, Benin Demographic and Health Survey; aOR, adjusted odds ratio; CI, confidence intervals.

## Background

In 2017, only 50% of newborn babies were breastfed within an hour and 60% were exclusively breastfed [1]. Early initiation of breastfeeding (EIBF) has been proven in studies to avert nearly a quarter of newborn fatalities worldwide in the last decade, however delaying initiation of breastfeeding beyond an hour increases the risk of neonatal mortality [1–3]. As a result, child survival has become a major public health priority in low- and middle-income countries (LMICS), particularly Sub-Saharan Africa (SSA) [4]. This major public health concern propelled the World Health Organization (WHO) in 2017 to recommend EIBF within the first hour of delivery to ensure that babies receive antibodies to prevent infections and chronic illnesses [5–7]. EIBF is a key child survival strategy that activates early milk let-down and ensures that the baby receives colostrum, which enhances immunity, growth and development [8].

Breastfeeding is one of the well-known and appropriate techniques for feeding newborn infants, reducing birth stress for the children and the mother as well as controlling children's temperature [9, 10]. EIBF is also associated with the reduction in the risk of breast and ovarian cancer as well as postpartum hemorrhage and obesity [11, 12].

In spite of the benefits associated with EIBF, a greater proportion of mothers in low-middle-income countries (LMICs) delay EIBF leading to higher risks of neonatal mortality and overall burden of disease [1, 8, 13]. The prevalence of EIBF in SSA is as low as 36.3% in Gambia and as high as 87.3% in Burundi [14]. Globally, there have been efforts to improve EIBF through the Sustainable Development Goals, the Innocenti Declaration as well as the Global Nutrition Targets 2025 which seek to increase global rates of exclusive breastfeeding and early initiation of breastfeeding to at least 50% by 2025 [15].

Realizing the importance of EIBF, the Government of Benin established a multi-sectoral, multi-stakeholder platforms and National Council for Food and Nutrition which has laid out both nutrition-specific and nutrition-sensitive approaches to improve EIBF [16]. Since 2017, the Government has participated in World Breastfeeding Week and has been committed to several pro-EIBF policies including the National Communication Strategy for Social and Behavioural Change; and the Community-Based Nutrition Interventions in order to increase the prevalence of EIBF [17].

Previous studies have suggested that women's employment status, educational level, parity, age, mode of delivery, place of delivery, wealth status, place of birth, antenatal care (ANC) visit, place of residence, mass media, community literacy level and community socioeconomic status are associated with EIBF [2, 18–22].

Studies in Benin have focused on socio-cultural beliefs [16], factors in rural South [23, 24], essential newborn care practices [25] and provision of support to improve breastfeeding practices [26]. Although the WHO and United Nation Children's Fund (UNICEF) Global Strategy for infant and young child feeding recommend three breastfeeding practices, little is known about the prevalence and determinants of EIBF in Benin at the individual, household and community levels. This study seeks to contribute to research related to EIBF in Benin by investigating the prevalence and determinants of EIBF in the country. Furthermore, the study will provide important evidence that can be used by policymakers and program managers to design and implement programs that will improve EIBF in Benin.

## Materials and methods

### Study area

Benin is a country in West Africa that is situated east to Togo and west of Nigeria. It is bordered to the north by Burkina Faso and Niger and the south by the Gulf of Guinea. Benin,

formerly known as the Republic of Dahomey, is officially called the Republic of Benin has a population of about 13 million on an area of 114,763 $Km^2$. The country has a fertility rate of 5.7 children per woman and a life expectancy of 61.2 years [27]. Infant mortality rate has witnessed a marginal decline from 64 to 55 deaths per 1,000 live births in the last decade [28].

## Data source and study population

The study made use of data from the most recent Benin Demographic and Health Surveys (BDHS) conducted in 2017/2018. This data was collected from November 2017 to February 2018 with a sample size of 15,928 women. The Demographic and Health Survey (DHS) is a countrywide representative study undertaken in a five-year period in several LMICs in Asia and Africa. It focuses on maternal and child health by interviewing women in their reproductive age (15–49 years). The DHS follows standardized procedures in areas such as sampling, questionnaires, data collection, cleaning, coding, and analyses, which allow for comparison across countries. Details of the methodology, instruments, pretesting of the instruments, training and recruitment of enumerators are documented in the final report of the 2017/2018 BDHS [29]. For this analysis, we used children's data set with a total weighted sample of 7,223 eligible women aged 15 to 49 years. We relied on the Strengthening the Reporting of Observational Studies in Epidemiology (STROBE) statement in drafting the manuscript [30]. The dataset is freely available for download at: https://dhsprogram.com/data/dataset/Benin_Standard-DHS_2017.cfm?flag=1.

## Description of variables

### Outcome variables

The outcome variable for the study was EIBF. EIBF has been defined as the initiation of breastfeeding within the first hour (1 hour) of birth [19, 31]. From BDHS, this variable was derived from the question "when was child put to breast?". The response options to this question were "immediately," "within first hour", "hours," and "days". For this study, the response options were recoded into "1" = EIBF (immediately and within first hour) and "0" = late breastfeeding initiation (hours and days). Several studies have used similar coding [8, 31–33].

### Independent variables

Based on literature, 17 variables were investigated as determinants of EIBF practice among women of reproductive age in Benin [31–35]. The variables were grouped as individual, households, and community-level factors. The individual-level variables included age (15–19, 20–24, 25–29, 30–34, 35–39, 40–44 and 45–49 years), education (no education, primary, and secondary/higher), partner's education (no education, primary, and secondary/higher), employment status (not working and working), parity (no birth, one birth, two births, three births, four or more births), number of ANC visits (no visit, less than 4, and 4 or more), place of delivery (home and health facility), sex of child (male and female), mode of delivery (vaginal and caesarean section), healthcare decision making-capacity (alone and not alone), assisted birth (skilled and unskilled) and media mass exposure. Listening to radio, watching television, and reading newspaper/magazine were coded as media exposure. These three variables had the same response options; "not at all", "less than once a week", and "at least once a week". Based on literature, we grouped the response options into "No" which meant no mass media exposure (not at all) and "Yes" to mean mass media exposure (less than once a week and at least once a week) [31, 32].

The household-level factors are ethnicity (Adja and related, Bariba and related, Dendi and related, Fon and related, Yoa, Lokpa and related, Betamarib and related, Peulh and related, Youraba and related and others.) and wealth (Poorest, poorer, middle, richer, and richest) while the community-level factors are place of residence (urban and rural), community literacy level (low, moderate, and high), and community socioeconomic status (low, moderate, and high) as categorized by previous studies [8, 31].

## Data analysis

STATA was used to analyze the data for this study. First, a pie chart was used to show the prevalence of EIBF practice in Benin (see Fig 1). This was followed by weighted frequencies and percentages of the independent variables. Bivariate analysis was used to present results of the distribution of EIBF across the independent variables using Pearson's Chi-square ($\chi^2$) to test for proportional differences (Table 1). Variables that showed statistical significance in the bivariate analysis were further moved to the multilevel logistic regression. The multilevel logistic regression analysis had fixed effects and random effects. Using the variance inflation factor (VIF), the multicollinearity test showed that there was no evidence of collinearity among the explanatory variables (mean VIF = 3.65).

In Table 2, there are 4 models. The four models show the nature of the independent variables which were categorized as individual level, household-level and community-level variables. This allows to see the effects of the relationship between the independent variables and the outcome variable. The first model (Model 0) was an empty model where no explanatory variable was used and the result showed the variance of EIBF attributable to the distribution of the primary sampling units. Model 1 took into account only individual level variables, while Model 2 had both the household-level and community-level variables. Model 3, which was the complete model, had both the individual, household-level and community-level variables. The results of the fixed effects were presented as adjusted odds ratios with their corresponding 95%

## Prevalence of early initiation of breastfeeding

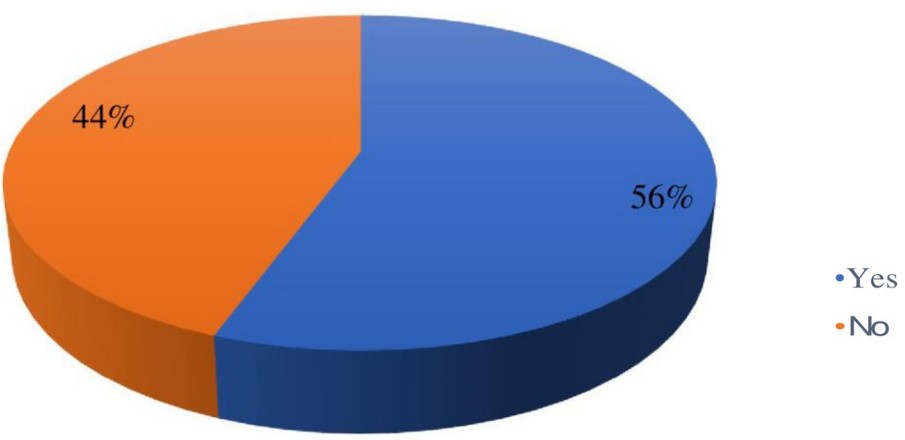

Source: 2017/8 Benin Demographic and Health Survey

**Fig 1. Prevalence of early initiation of breastfeeding.**

**Table 1. Distribution of EIBF across independent factors (weighted, N = 7,223).**

| Variables | Frequency (%) | Weighted (%) | EIBF (%) | Chi-square, p-value |
|---|---|---|---|---|
| **Age** | | | | $\chi^2 = 5.01$, p = 0.534 |
| 15–19 | 361 | 5.0 | 56.7 | |
| 20–24 | 1,494 | 20.7 | 53.5 | |
| 25–29 | 2,088 | 28.9 | 55.6 | |
| 30–34 | 1,506 | 20.9 | 56.8 | |
| 35–39 | 1,082 | 15.0 | 55.4 | |
| 40–44 | 494 | 6.8 | 55.1 | |
| 45–49 | 198 | 2.7 | 61.8 | |
| **Education** | | | | $\chi^2 = 38.0$, p<0.001* |
| No education | 4,750 | 65.8 | 58.2 | |
| Primary | 1,259 | 17.4 | 50.9 | |
| Secondary+ | 1,214 | 16.8 | 49.9 | |
| **Partner's Education** | | | | $\chi^2 = 47.2$, p<0.0001* |
| No education | 3,938 | 54.5 | 58.7 | |
| Primary | 1,480 | 20.5 | 53.4 | |
| Secondary+ | 1,805 | 25.0 | 50.5 | |
| **Employment status** | | | | $\chi^2 = 17.1$, p<0.0001* |
| Not working | 1,203 | 16.7 | 60.9 | |
| Working | 6,020 | 83.3 | 54.5 | |
| **Parity** | | | | $\chi^2 = 15.8$, p<0.001* |
| One birth | 1,252 | 17.4 | 50.4 | |
| Two births | 1,343 | 18.6 | 54.0 | |
| Three births | 1,245 | 17.2 | 55.9 | |
| Four or more births | 3,383 | 46.8 | 58.0 | |
| **Number of ANC visit** | | | | $\chi^2 = 21.0$, p<0.001* |
| No visit | 791 | 11.0 | 59.7 | |
| Less than 4 | 2,582 | 35.7 | 57.5 | |
| 4 or more | 3,850 | 53.3 | 53.4 | |
| **Place of Delivery** | | | | $\chi^2 = 12.5$, p<0.001* |
| Home | 1.045 | 14.5 | 60.9 | |
| Health facility | 6,178 | 85.5 | 54.7 | |
| **Mass media exposure** | | | | $\chi^2 = 32.7$, p<0.0001* |
| No | 2,784 | 38.5 | 59.4 | |
| Yes | 4,439 | 61.5 | 53.1 | |
| **Sex of child** | | | | $\chi^2 = 0.0$, p = 0.912 |
| Male | 3,624 | 50.2 | 55.5 | |
| Female | 3,599 | 49.8 | 55.7 | |
| **Modes of delivery** | | | | $\chi^2 = 168.2$, p<0.001* |
| Vaginal | 6,811 | 94.3 | 57.4 | |
| Caesarean section | 412 | 5.7 | 25.5 | |
| **Healthcare decision making capacity** | | | | $\chi^2 = 2.6$, p = 0.107 |
| Alone | 668 | 9.3 | 60.4 | |
| Not alone | 6,555 | 90.7 | 55.1 | |
| **Assistant at Birth** | | | | $\chi^2 = 44.1$, p<0.001* |
| Skilled | 5,795 | 19.8 | 64.5 | |
| Unskilled | 1,428 | 80.2 | 53.4 | |
| **Ethnicity** | | | | $\chi^2 = 25$, p<0.001* |
| Adja and related | 909 | 12.6 | 52.7 | |
| Bariba and related | 903 | 12.5 | 57.6 | |

*(Continued)*

**Table 1.** (Continued)

| Variables | Frequency (%) | Weighted (%) | EIBF (%) | Chi-square, p-value |
|---|---|---|---|---|
| Dendi and related | 455 | 6.3 | 76.1 | |
| Fon and related | 2,400 | 33.2 | 53.0 | |
| Yoa, Lokpa and related | 242 | 3.4 | 68.6 | |
| Betamarib and related | 454 | 6.2 | 62.6 | |
| Peulh and related | 748 | 10.4 | 58.5 | |
| Youraba and related | 722 | 10.0 | 35.9 | |
| Others | 390 | 5.4 | 63.7 | |
| **Wealth** | | | | $\chi^2 = 25.7$, p$<$0.001* |
| Poorest | 1,530 | 21.2 | 57.4 | |
| Poorer | 1,456 | 20.1 | 59.5 | |
| Middle | 1,487 | 20.6 | 55.4 | |
| Richer | 1,450 | 20.1 | 53.1 | |
| Richest | 1,300 | 18.0 | 52.1 | |
| **Place of residence** | | | | $\chi^2 = 6.8$, p = 0.009* |
| Urban | 2,752 | 38.1 | 53.1 | |
| Rural | 4,471 | 61.9 | 57.1 | |
| **Community literacy level** | | | | $\chi^2 = 53.2$, p$<$0.001* |
| Low | 2,961 | 41.0 | 60.0 | |
| Medium | 2,511 | 34.8 | 54.2 | |
| High | 1,751 | 24.2 | 50.0 | |
| **Community socioeconomic status** | | | | $\chi^2 = 29.2$, p$<$0.001* |
| Low | 3,498 | 48.4 | 58.4 | |
| Moderate | 1,879 | 26.0 | 53.1 | |
| High | 1,846 | 25.6 | 52.7 | |

*p-value less than 0.05 indicates statistical significance

confidence intervals signifying their level of precision. The random effects were assessed with Intra-Cluster Correlation (ICC) while log-likelihood ratio (LLR) and Akaike's Information Criterion (AIC) tests were used for the model comparisons. The ICC is calculated as a ratio ICC = (variance of interest) / (total variance) = (Variance of interest) / (variance of interest + unwanted variance) [36]. The STATA command 'melogit' was used in fitting these models. Model comparison was done using the log-likelihood ratio (LLR) and the Akaike's Information Criterion (AIC) tests. The complex nature of the sampling structure of the data was adjusted using the STATA Survey command 'svyset v021 [pweight = wt], strata (v023)'.

## Ethics statement

Authors requested for the BDHS dataset (https://www.dhsprogram.com/) and obtained permission to access and download the data files for research purpose only. Further information about Benin DHS data usage and ethical standards are available at: https://dhsprogram.com/publications/publication-FR350-DHS-Final-Reports.cfm.

## Results

### Prevalence of Early Initiation of Breastfeeding

From the study, the prevalence of EIBF among mothers was 56% (95% CI = 0.54–0.57) (see Fig 1).

**Table 2. Multilevel logistic regression analysis on predictors of EIBF in Benin.**

| Variables | Model 0 | Model 1 aOR (95% CI) | Model 2 aOR (95% CI) | Model 3 aOR (95% CI) |
|---|---|---|---|---|
| **Fixed effect** | | | | |
| *Individual level factors* | | | | |
| **Education** | | | | |
| No education | | 1 | | 1 |
| Primary | | 0.87 (0.74–1.02) | | 0.88 (0.75–1.03) |
| Secondary+ | | 0.92 (0.77–1.11) | | 0.95 (0.79–1.14) |
| **Partner's Education** | | | | |
| No education | | 1 | | |
| Primary | | 1.02 (0.87–1.19) | | 1.05 (0.90–1.23) |
| Secondary+ | | 0.97 (0.82–1.14) | | 0.98 (0.83–1.16) |
| **Employment status** | | | | |
| Not working | | 1 | | 1 |
| Working | | 0.78*** (0.67–0.91) | | 0.80** (0.69–0.94) |
| **Parity** | | | | |
| One birth | | 1 | | 1 |
| Two births | | 1.21* (1.01–1.45) | | 1.22* (1.02–1.46) |
| Three births | | 1.23* (1.02–1.47) | | 1.25* (1.04–1.51) |
| Four or more births | | 1.30** (1.10–1.52) | | 1.33*** (1.14–1.57) |
| **Number of ANC visit** | | | | |
| No visit | | 1 | | 1 |
| Less than 4 | | 1.14 (0.91–1.43) | | 1.13 (0.90–1.41) |
| 4 or more | | 1.00 (0.80–1.26) | | 1.01 (0.79–1.27) |
| **Place of Delivery** | | | | |
| Home | | 1 | | 1 |
| Health facility | | 1.69*** (1.29–2.21) | | 1.75*** (1.33–2.28) |
| **Mass media** | | | | |
| No | | 1 | | 1 |
| **Yes** | | 0.84** (0.74–0.95) | | 0.85** (0.75–0.96) |
| Modes of delivery | | | | |
| Vaginal | | 1 | | 1 |
| Cesarian section | | 0.21*** (0.17–0.28) | | 0.21*** (0.16–0.28) |
| **Assistant at Birth** | | | | |
| Unskilled | | 1 | | 1 |
| Skilled health professionals | | 0.56*** (0.45–0.70) | | 0.57*** (0.46–0.71) |
| *Household level factors* | | | | |
| **Ethnicity** | | | | |
| Adja and related | | | 1 | 1 |
| Bariba and related | | | 1.14 (0.84–1.54) | 1.06 (0.77–1.45) |
| Dendi and related | | | 2.40*** (1.64–3.52) | 2.10*** (1.42–3.12) |
| Fon and related | | | 0.97 (0.77–1.22) | 1.02 (0.80–1.29) |
| Yoa, Lokpa and related | | | 1.61* (1.05–2.46) | 1.62* (1.04–2.53) |
| Betamarib and related | | | 1.60* (1.11–2.30) | 1.53* (1.06–2.23) |
| Peulh and related | | | 1.29 (0.94–1.78) | 1.27 (0.91–1.78) |
| Youraba and related | | | 0.64** (0.48–0.86) | 0.62** (0.46–0.85) |
| Others | | | 0.98 (0.70–1.37) | 0.88 (0.62–1.25) |
| **Wealth** | | | | |

*(Continued)*

**Table 2.** (Continued)

| Variables | Model 0 | Model 1 aOR (95% CI) | Model 2 aOR (95% CI) | Model 3 aOR (95% CI) |
|---|---|---|---|---|
| Poorest | | | 1 | 1 |
| Poorer | | | 1.20* (1.00–1.43) | 1.22* (1.02–1.46) |
| Middle | | | 1.13 (0.94–1.36) | 1.18 (0.97–1.43) |
| Richer | | | 1.06 (0.86–1.30) | 1.16 (0.93–1.44) |
| Richest | | | 1.14 (0.88–1.47) | 1.42* (1.08–1.87) |
| *Community level-factors* | | | | |
| **Place of residence** | | | | |
| Urban | | | 1 | 1 |
| Rural | | | 1.05 (0.84–1.31) | 1.03 (0.82–1.30) |
| **Community literacy level** | | | | |
| Low | | | 1 | 1 |
| Medium | | | 0.86 (0.67–1.10) | 0.88 (0.68–1.13) |
| High | | | 0.68 (0.49–0.94) | 0.73 (0.52–1.03) |
| **Community socioeconomic status** | | | | |
| Low | | | 1 | 1 |
| Moderate | | | 0.92 (0.71–1.19) | 0.94 (0.73–1.22) |
| High | | | 1.01 (0.71–1.44) | 1.06 (0.74–1.52) |
| **Random effect results** | | | | |
| PSU variance (95% CI) | 0.94 (0.77–1.15) | 0.93 (0.76–1.13) | 0.74 (0.60–0.92) | 0.79 (0.63–0.98) |
| ICC | 22.3% | 22.0% | 18.4% | 19.3% |
| LR Test | $\chi^2 = 593.52$, p<0.001 | $\chi^2 = 558.03$, p<0.001 | $\chi^2 = 383.70$, p<0.001 | $\chi^2 = 400.58.$, p<0.001 |
| Wald chi-square | Ref | 210.13 | 84.32 | 272.04 |
| **Model fitness** | | | | |
| Log-likelihood | -4670.31 | -4551.57 | -4629.74 | -4519.54 |
| AIC | 9344.62 | 93135.14 | 9297.48 | 9105.08 |
| **N** | 7,223 | 7,223 | 7,223 | 7,223 |

Source: 2017/2018 Benin Demographic and health Survey

*$p < 0.05$

**$p < 0.01$

***$p < 0.001$, aOR adjusted odds ratio

PSU Primary Sampling Unit; ICC Intra-Class Correlation; aOR adjusted odds ratio; LR Test Likelihood Ratio Test; AIC Akaike's Information Criterion

## Distribution of EIBF across the independent variables

The results showed that the largest proportion of the respondents (28.9%) were aged 25–29. Almost two-thirds of the respondents (65.8%) had no formal education while more than half of their partners (54.5%) had no formal education. The majority (83.3%) of the women were belonged to the working class. Almost half (46.8%) had four or more births, and a little above half (53.3%) of women had four or more antenatal care (ANC) visits. A higher proportion (85.5%) of women had their delivery at a health facility, whilst 6 out of 10 (61.3%) were exposed to mass media. Most of the respondents (80.2%) were attended to by skilled health professionals, 1 out of 2 of the participants (50.2%) children were males and over 90% (94.3%) of the women had vaginal delivery. With regards to healthcare decision making, almost all of the respondents (90.7%) indicated that they do not make decision alone and about one-third of them (33.2%) were from Fon and related ethnic group. The highest proportion of the women (21.2%) were in the poorest wealth quintile, whereas more than half (61.9%) resided in rural areas. The majority of the women

(41.0%) within low community literacy level while the highest proportion of the women (48.4%) were within low community socioeconomic status (See Table 1).

The result shows that the highest prevalence of EIBF was among women aged 45–49 (61.8%), women who had no formal education (58.2%), women whose partners had no formal education (58.7%) and not working women (60.9%). Similarly, women with four births (58.0%), women with no ANC visit (59.7%), women with home delivery (60.9%), women who had no mass media exposure (59.4%), women with female child (55.7%), women who had vaginal delivery (57.4%) and women who made healthcare decisions alone (60.4%) had the highest prevalence of EIBF. With regards to birth assistance, women who had skilled health professionals' assistance during delivery (64.5%), women affiliated with Dendi and related ethnic group (76.1%), who were in the poorer wealth quintile (59.5%), women who resided in the rural areas (57.1%), women were in low community literacy level (60.0%) and women were in low community socioeconomic status (58.4%) had the highest prevalence of EIBF (Table 1).

### Multilevel logistic regression analysis on predictors of EIBF in Benin

Model 3 of Table 2 illustrates the results of the multilevel logistic regression analysis on the predictors of EIBF in Benin. Women who were working had reduced odds of EIBF (aOR = 0.80, 95% CI = 0.69–0.94) relative to those who were not working. Parity showed a direct relationship with the probability of practicing EIBF. Women with four or more births (aOR = 1.33, 95% CI = 1.14–1.57) had higher odds of EBF compared with those with one birth. Women who had delivered at health facilities had higher odds of breastfeeding their children within an hour after birth (aOR = 1.75, 95% CI = 1.33–2.28) than their counterparts who delivered at home. Women who were exposed to mass media had lower odds of EIBF practice (aOR = 0.85, 95% CI = 0.75–0.96) than women who were not exposed to mass media.

The study's findings revealed that women who had caesarean section had lower odds of EIBF practice (aOR = 0.21, 95% CI = 0.16–0.28) than their counterparts who had vaginal delivery. Regarding birth assistance, women who received assistance at birth from skilled workers had lower odds of EIBF practice (aOR = 0.57, 95% CI = 0.46–0.71) than their counterparts who received assistance at birth from unskilled worker. Compared with Adja and related ethnic group, women of Dendi and related ethnic group had increased odds of EIBF practice (aOR = 2.10, 95% CI = 1.42–3.12). On the other hand, women in the richest wealth quintile had higher odds breastfeeding their children within an hour after birth (aOR = 1.42, 95% CI = 1.08–1.87) compared with poorest women (See Table 2).

### Measures of variation (random effects)

The ICC value for the Model 0 shows that 22.3% of the variation in EIBF practice was attributed to the between clusters variance. The variation between clusters then decrease to 22.0% in Model 1 which was the individual-level only model. The ICC further decreased to 18.4% in Model 2 which had both the household-level and community-level factors only model, but increased to 19.3% in Model 3 which is the complete model. This can be attributed to the differences in the clustering of the PSUs account for the variations in EIBF practice. From the model specification analysis, Model 3 which is the complete model with individual, household-level and community-level factors had the lowest AIC compared to the other models affirming the goodness of fit (see Table 2).

## Discussion

Although early initiation of breastfeeding is not only the simplest, it has also been considered as the most cost-effective and [18, 37]. According to literature, EIBF is one of the best

investments for increasing human capital, stimulating economic growth, and ensuring that all children have equitable access to opportunity. Hence, breastfeeding has been noted to have at least contributed more than $300 billion to the world economy each year [38, 39]. This study aimed to examine the prevalence and determinants of EIBF among women of reproductive age in Benin. Despite the fact that the government of Benin has undertaken many initiatives to improve EIBF, including Nutrition at the Centre Phase 1 and 2, Community Nutrition Programme, and Multisectoral Food, Health, and Nutrition Plan [16], the prevalence of EIBF from this study was 56.0%. This means that not all babies are breastfed within an hour of delivery, which falls short of the WHO and UNICEF recommendation of 90%. The prevalence found in this study is lower than prevalence of 58.3% in SSA [19], 83.7% in Egypt [40], 83.1% in rural part of west Ethiopia [41] and 66.4% in Nepal [42]. However, it is higher than other prevalence of 45% reported in Nigeria [43], 36.4% in India [21], 36% in rural Bangladesh [44], and 8.5% in Pakistan [45]. The low prevalence of EIBF in Benin could be due to inadequate health facilities and healthcare service utilization in Benin, access to EIBF information, socio-economic status, and cultural difference [19, 22, 46].

From the multivariate logistic analysis, a significant association was found between EIBF and respondent's employment status, parity, mass media, modes of delivery, place of delivery, assistance at birth, and ethnicity.

Congruent with studies in Benin [23], SSA [19, 22, 41] and Nepal [42], women who were working had a reduced probability of EIBF practice as compared to those who are not working. The reason could be that working women may not be interested to start breastfeeding because of possibility of discontinuation of breastfeeding due to work. Another reason could be that working women may be financial stable to afford infant formula. Also, the non-existence of safe breastfeeding room at workplace, difficulties in expressing and storing milk, short maternal leave period, none-existence paternity leave and less availability of remote work could be the reason for the significant difference. There is therefore the need for employers to consider providing nursery room and storage facilities for mothers to breastfeed infants. In addition, organizations should consider allowing mothers work remotely where possible especially during the breastfeeding period and there is the need for policy makers to review the current maternity. Therefore, there is the need to explore the impact of employment and EIBF in Benin.

Similarly, early initiation of breastfeeding is significantly associated with parity. In this present study, women with four or more births had higher odds of EIBF compared with those having single birth. The finding concurs with some previous studies across SSA [3, 47, 48], Saudi Arabia [49] and Nepal [42]. This evidence suggests that women with higher parity may have knowledge and experience of the benefits associated with EIBF.

Furthermore, the finding from this study reveals that there is a significant relationship between early initiation of breastfeeding and place of delivery. Women who delivered at health facilities had higher odds of EIBF. This finding corroborates with some previous literature conduced in SSA [19, 50, 51]. The plausible reason could be that women who delivered at health facilities may be supported and made aware of the importance of EIBF. Also, women who delivered at home practice prelacteal feeding and this may result in delayed initiation of breastfeeding. Given the importance of EIBF and place of delivery, stakeholders should place more emphasis on health facilities delivery during ANC visit of women.

Women with exposure to mass media had lower odds of early initiation to breastfeeding compared to women who had no exposure to mass media. A study conducted in SSA was in agreement with the finding of this present study [19]. However, studies conducted in Ethiopia were in contrast with this particular finding of this study [41, 52]. This might be due to Benin's mass media not advocating early breastfeeding initiation and instead promoting infant formula feedings, milk substitutes, teats, and bottles.

Congruent with previous studies conducted in rural southern Benin [23], southern Ethiopia [53], Nigeria [54], and Ghana [55], ethnicity was significantly associated with EIBF. Women from Dendi and related ethnic group were more likely to initiate breastfeeding with one hour. The possible explanation is the sociocultural beliefs and geographical location coupled with low socioeconomic status influence early initiation of breastfeeding. Therefore, increased odds of early initiation of breastfeeding among women from Dendi and related ethnic group could be the acceptability and the knowledge of the benefits associated with EIBF.

With respect to modes of delivery, women who had caesarean section were less likely than those who had vaginal delivery to practice EIBF. The finding of this study is consistent with some studies in SSA [49, 56, 57] and Saudi Arabia [58]. The plausible reason for this could be that women who have caesarean section could experience discomfort, fatigue, pain, and anaesthesia effects after the procedure [22, 46, 59]. Caesarean section is a lifesaving tool for the mother and the newborn; however, this could make it difficult to practice EIBF [60]. Therefore, there is the need to institute appropriate interventions for mothers who have undergone caesarean section deliveries to initiate breastfeeding their children within an hour after birth. Healthcare facilities can implement a strategy to guarantee that babies delivered through caesarean section are breastfed promptly, either in the operating room or recovery area. Nonetheless, medical personnel in the delivery room should have the utmost authority to prohibit early initiation of breastfeeding if the mother's or baby's health will be jeopardized [61]. Also, caesarean sections should not be encouraged when it has no medical indications [62].

In relation to birth assistance, there was an association between birth assistance and early initiation of breastfeeding. Inconsistent with studies conducted in Ethiopia [63], and Zimbabwe [53], this study revealed that women who were attended to by skilled health professionals were less likely to initiate early breastfeeding. This might be due to residual confounding, as studies have shown that women who had their births attended by skilled health professionals are more likely to initiate early breastfeeding [54].

## Strengths and limitations

This study used the most current nationally representative data of the Benin Demographic and Health Survey in examining the prevalence and determinants of EIBF. The methods employed in sampling and data collection also support the representativeness of the study. Therefore, the findings and recommendations can be applied to all women in Benin and other low- and middle-income countries in Africa. With regards to the limitations, the study did not consider maternal knowledge on the benefits of EIBF because it was not captured in the dataset. Also, the cross-sectional nature of this study does not allow for causality to be inferred from the findings. Furthermore, the data was collected retrospectively and this could have led to some form of recall bias resulting in either over or under reporting of EBF.

## Conclusion and implication

The findings of this study showed that four in ten children miss EIBF in Benin. The study revealed that employment status, parity, mass media, modes of delivery, place of delivery, assistance at birth and ethnicity were significantly associated with EIBF. Therefore, policymakers need to consider these factors when developing policies to promote and enhance EIBF in Benin. It is further recommended that information, education and communication programs targeting mothers who are less likely to practice EIBF be formulated, implemented and monitored accordingly by the Ministry of Health and its agencies. Also, there is the need for the Ministries of Health, Higher Education and Scientific Research, and Secondary, Technical and

Vocational Education to embark on community outreach programs on the importance of EIBF practice.

## Acknowledgments

The authors are grateful to MEASURE DHS for granting access to the datasets used in this study. Also, authors would like to thank Dr. Edward Kwabena Ameyaw for the technical support and revisiting the language of this manuscript.

## Author Contributions

**Conceptualization:** Ebenezer Kwesi Armah-Ansah.

**Data curation:** Ebenezer Kwesi Armah-Ansah.

**Formal analysis:** Ebenezer Kwesi Armah-Ansah.

**Methodology:** Ebenezer Kwesi Armah-Ansah.

**Supervision:** Ebenezer Kwesi Armah-Ansah.

**Writing – original draft:** Ebenezer Kwesi Armah-Ansah, Elvis Ato Wilson, Kenneth Fosu Oteng, Benedicta Bawa, Joseph Yaw Dawson.

**Writing – review & editing:** Ebenezer Kwesi Armah-Ansah, Elvis Ato Wilson, Kenneth Fosu Oteng, Benedicta Bawa, Joseph Yaw Dawson.

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
