## [Decision Letter · Decision Letter 0]

20 Mar 2023

PGPH-D-23-00087

Examining the prevalence and determinants of early initiation of breastfeeding: Evidence from the 2018 Benin Demographic and Health Survey

Dear Dr. Armah-Ansah,

Thank you for submitting your manuscript to PLOS Global Public Health. After careful consideration, we feel that it has merit but does not fully meet PLOS Global Public Health’s publication criteria as it currently stands. Therefore, we invite you to submit a revised version of the manuscript that addresses the points raised during the review process.

While I find your article to be of significant interest, there are several limitations as highlighted by the reviewers that detract from it. Kindly review each comment and address it adequately. 

We look forward to receiving your revised manuscript.

Kind regards,

Godfred Boateng

Academic Editor

Reviewers' comments:

Reviewer's Responses to Questions

**Comments to the Author**

1. Does this manuscript meet PLOS Global Public Health’s publication criteria? Is the manuscript technically sound, and do the data support the conclusions? The manuscript must describe methodologically and ethically rigorous research with conclusions that are appropriately drawn based on the data presented.

Reviewer #1: Yes

Reviewer #2: Yes

2. Has the statistical analysis been performed appropriately and rigorously?

Reviewer #1: Yes

Reviewer #2: Yes

3. Have the authors made all data underlying the findings in their manuscript fully available (please refer to the Data Availability Statement at the start of the manuscript PDF file)?

Reviewer #1: Yes

Reviewer #2: Yes

4. Is the manuscript presented in an intelligible fashion and written in standard English?

Reviewer #1: No

Reviewer #2: Yes

5. Review Comments to the Author

Reviewer #1: Review Comments

Thank you for the opportunity to review this paper. Though an important topic, there are several issues the authors should address before the paper can be considered for publication.

Comments/ Suggestions

Abstract

1 Avoid abbreviations in the abstract section

2 Method should revised and edited?

3 Conclusion: it could be revised hence you stated recommendations without some presentation of results? Before recommendation you should summarize results and provide suggestions to the concerned body?

Background

Grammatical error and some paragraphs are poorly English written. Generally revision should be needed.

Methods

Line 163-164 is confusing rewrite in clear way or you should remove it.

What could be your justification to use LLR and AIC for model comparison; is there any conditions to use these interchangeably for model comparison? How?

Result

Line 182 presented about the overall prevalence of EIBF i.e. point estimation. You must mention the interval estimation of the prevalence.

How do you measure media exposure?

In the results section you mention model building i.e. measure of variation ICC …. But I did not see anything about that in the methods part. The result should be in line with methods part? You should mention the mathematical formula how to compute ICC and others?

Discussion

Line 259-260 the sentence should be inconvenience should remove it or rewrite in clear cut way.

The justification regarding the prevalence is poorly explain please improve it by accounting the specific reasons found at hand in Benin?

Line 273-276 this paragraph was implication of the study but the implication of the study should mention the recommendation parts. Be focus on the nationalities of the study and justify it more about your findings with other studies published elsewhere.

Your discussion focusing only USA and Nepal why? You should widening your eye to see the variation and similarity of findings published in Africa and elsewhere? Hence you get full insight if you compare similar country published on it.

Generally your discussion needs extensive revision hence the current discussion is shallow, you explain in short and clear cut way.

Conclusion

Line 328-329 not convincing recommendation? The recommendation should be mention in specific and targeted way.

Please minimize your recommendation hence the current one is too long and difficult to understand? As stated before the recommendation should be short, clear and specific to the targeted authority and individual?

Reviewer #2: Overview:

The primary purpose of this study was to examine the prevalence and determinants of early initiation of breastfeeding in Benin. This topic is interesting and relevant, given the far-reaching implications early initiation of breastfeeding can have on the health and wellbeing of newborn infants and mothers. However, major revisions are needed to strengthen the ideas in the paper.

Comments

Abstract:

1.    The first sentence in the abstract seems to suggest that early initiation of breastfeeding is beneficial only in low- and middle-income countries and not in other contexts. Is that what the authors are implying? If not, please rephrase the sentence to suggest that regardless of context, early initiation of breastfeeding is crucial for avoiding early childhood fatalities.

2.    It is not relevant to indicate which version of Stata was used for the analysis in the result section of the abstract.

3.    Several of the variables listed in the conclusion section of the abstract are not statistically significant and should be excluded.

4.    Also, it is not clear what the authors mean by “health professionals must minimize indications of caesarean delivery.”

Introduction:

1.     There is no clear connection between the ideas in the first two sentences in the background section of the paper. The authors noted that “Globally, in 2017, only 50% of newborn babies were breastfed within an hour and 60% of them were exclusively breastfed [1]. To improve logical flow, it would be great if the authors could briefly explain why this is the case and link this to discussions around child survival before they highlight Child survival is a major public health concern in low- and middle-income countries.

2.    On line 82, the authors note that “As of 2016, EIBF had prevented more than twenty-thousand breast cancer-related deaths [11].” I recommend the authors remove this sentence. They cannot make this claim based on a cross-sectional study conducted in rural Niger.

3.    They also note that “EIBF has prevented almost a quarter of neonatal deaths in the last decade [1, 11 (see line 84).” Is this globally or in low- to middle-income countries? Kindly present some statistics to support this claim.

4.    The authors note that “This study seeks to contribute to research related to EIBF in Benin by investigating the prevalence and determinants of EIBF in the country (see line 103).” Which determinants are they focusing on and why? They need to include a review of the key literature that informed the selection of their variables of interest and explain why we should expect them to have an effect on EIBF.

Materials and Methods

1.    It is not relevant to specify what the capital city of Benin is (see line 113); rather, the emphasis should be on what the child survival rate in Benin is. Is EIBF an issue in Benin? These are useful to justify why Benin is an important study context.

2.    A detailed description of your independent variables is required. Perhaps you can construct a table with the following required information: variables, question asked, response options, and coding strategy.

3.    It is not relevant to indicate which Stata commands were used for the analysis (see lines 175 and 178)

Results

1.    Authors are encouraged to interpret the descriptive results and not just report the percentages. For example, the authors report this: “The results showed that 28.9% of the respondents were aged 25-29; 65.8% had no formal education; partners of 54.5% had no formal education whilst 83.3% of the participants were working.”

This can be written as follows: A large share of the respondents (70%) were between the ages of 20 and 34.

More than 4 out of 5 of them (83%) were employed; however, most of the respondents (65%) had no formal education. The share of spouses without formal education is also quite high (54%).

2.    The authors estimated 3 separate models but only interpreted Model 3. It is surprising why they did not interpret the remaining 2 models, and I would encourage them to do so, at least briefly. It would also be helpful for the authors to explain why they estimated 3 separate models.

3.    Some of the results are not statistically significant (e.g., education and partner’s education), but the authors discussed them as though they were. Only variables that are statistically significant should be discussed in detail. The authors are also encouraged to ensure they are reporting the right confidence intervals.

Discussion

1.    It is not clear what the authors mean by “Early initiation of breastfeeding has been considered as the cost effective .... (Line 254)." Cost effective in what sense? Further clarification is required.

2.    On lines 258 to 260, the authors noted that “This prevalence is lower than those reported by existing studies in SSA [32, 33, 34] and Nepal [35]. However, it is lower than what has been reported by other studies in Asia [36, 37, 38]. What prevalence rate has been reported for SSA, Nepal and Asia?

3.    The authors should focus their discussion on variables that are statistically significant.

4.    On lines 302, the authors note that “there is the need to institute appropriate interventions for mothers who have undergone caesarean section deliveries to initiate breastfeeding their children within an hour after birth." It would be insightful for the authors to briefly clarify what could be considered an appropriate intervention in this case.

5.    The explanation for the surprising finding that women who were attended to by skilled health professionals were less likely to initiate early breastfeeding (see lines 308 to 310) is not convincing enough and requires further elaboration and clarification. The authors seem to suggest that women who are attended to by skilled health professionals are more likely to experience complications than women who are not, which is quite the opposite. Moreover, cesarean section should not be considered as a possible explanation since it was included in the analysis as a control variable.

6.    On line 317, the authors note that the study did not consider maternal beliefs. Why is this important to consider?

7.    The authors suggest that health professionals must minimize indications of caesarean delivery, setup systems that will enable skin to-skin and establish practices that enable early initiation of breastfeeding among women who have undergone caesarean section (lines 330 to 333). It is not quite clear what the authors mean, and further clarification and elaboration are require.

6. PLOS authors have the option to publish the peer review history of their article (what does this mean?). If published, this will include your full peer review and any attached files.

**Do you want your identity to be public for this peer review?** For information about this choice, including consent withdrawal, please see our Privacy Policy.

Reviewer #1: No

Reviewer #2: No

---

## [Decision Letter · Decision Letter 1]

19 Jul 2023

Examining the prevalence and determinants of early initiation of breastfeeding: Evidence from the 2017/2018 Benin demographic and health survey

PGPH-D-23-00087R1

Dear Mr Armah-Ansah,

We are pleased to inform you that your manuscript 'Examining the prevalence and determinants of early initiation of breastfeeding: Evidence from the 2017/2018 Benin demographic and health survey' has been provisionally accepted for publication in PLOS Global Public Health.

Best regards,

Julia Robinson

Staff Editor

Reviewer Comments (if any, and for reference):

Reviewer's Responses to Questions

**Comments to the Author**

1. If the authors have adequately addressed your comments raised in a previous round of review and you feel that this manuscript is now acceptable for publication, you may indicate that here to bypass the “Comments to the Author” section, enter your conflict of interest statement in the “Confidential to Editor” section, and submit your "Accept" recommendation.

Reviewer #1: All comments have been addressed

2. Does this manuscript meet PLOS Global Public Health’s publication criteria? Is the manuscript technically sound, and do the data support the conclusions? The manuscript must describe methodologically and ethically rigorous research with conclusions that are appropriately drawn based on the data presented.

Reviewer #1: Yes

3. Has the statistical analysis been performed appropriately and rigorously?

Reviewer #1: Yes

4. Have the authors made all data underlying the findings in their manuscript fully available (please refer to the Data Availability Statement at the start of the manuscript PDF file)?

Reviewer #1: Yes

5. Is the manuscript presented in an intelligible fashion and written in standard English?

Reviewer #1: Yes

6. Review Comments to the Author

Reviewer #1: Good job, all the questions and suggestions are already addressed.

7. PLOS authors have the option to publish the peer review history of their article (what does this mean?). If published, this will include your full peer review and any attached files.

**Do you want your identity to be public for this peer review?** For information about this choice, including consent withdrawal, please see our Privacy Policy.

Reviewer #1: No
